# Skewed perception of personal behaviour as a contributor to antibiotic resistance and underestimation of the risks

**Emily Smith, Sarah Buchan**◉*

Department of Life and Environmental Science, Bournemouth University, Poole, United Kingdom

* sbuchan@bournemouth.ac.uk

## Abstract

The increasing prevalence of antibiotic-resistant bacteria poses a significant threat to global human health. Countering this threat requires the public to understand the causes of, and risks posed by, antibiotic resistance (AR) to support changing healthcare and societal approaches to antibiotic use. To gauge public knowledge, we designed a questionnaire to assess awareness of causes of AR (both personal and societal) and knowledge of absolute and relative risks posed by antibiotic-resistant bacteria. Our findings reveal that while >90% respondents recognized personal behaviours as limiting AR, few individuals recognized the importance of societal factors e.g. the use of antibiotics in livestock. Furthermore, more respondents named viruses (either by name or as a group) than bacteria as reasons to take antibiotics, indicating lack of understanding. The absolute numbers of current and predicted future deaths attributed to antibiotic-resistant bacteria were under-estimated and respondents were more concerned about climate change and cancer than AR across all age groups and educational backgrounds. Our data reveal that despite heightened public awareness of infection-control measures following the COVID-19 pandemic, there remains a knowledge gap related to contributors and impacts of increasing numbers of antibiotic-resistant bacteria.

## Introduction

Antibiotics have revolutionized patient care since their introduction in the early to mid 20<sup>th</sup> century yet the rise of antibiotic resistant and multi-drug-resistant bacteria is now a major global public concern [1]. Antibiotic resistant bacteria are responsible for approximately 700,000 deaths per year, a figure that is projected to increase to between 10 and 444 million deaths per year by 2050 [2–4]. Some have likened the scale of the challenge to that posed by climate change [5].

Over-prescription of antibiotics is widely recognized as a major contributor to the emergence of new antibiotic-resistant strains [6,7], with approximately 50% of anti-microbials prescribed thought to be unnecessary [1]. Furthermore, a recent study showed that around 70% of patients diagnosed with COVID-19 between January and April 2020 received a course of

**Data Availability Statement:** Raw data files are available through Bournemouth University's BORDaR database at https://doi.org/10.18746/bmth.data.00000316.

**Funding:** The authors received no specific funding for this work.

**Competing interests:** The authors have declared that no competing interests exist.

antibiotics, when only 7% of these patients had a bacterial co-infection [8]. A separate study in 2021 revealed that one fifth of Australians questioned took antibiotics to protect against COVID-19 [9]. Such marked misuse of antibiotics in the context of COVID-19 is predicted to further exacerbate the growth of antibiotic resistance (AR) worldwide [10].

The rise in prevalence of antibiotic-resistant bacteria can be attributed to multiple factors in addition to over-prescription. Some of these factors relate to personal behaviour/circumstances including sharing antibiotics with friends or family and ineffective disposal of unused medications [11,12]. Other factors are linked to societal behaviour, for example extensive use of antibiotics in livestock, global human migration, wildlife spread, poor sewerage and absence of new antibiotic discoveries [2,11]. Several campaigns have sought to educate the public about the factors that contribute to AR and to the risks posed by the emergence of antibiotic-resistant bacterial strains. Despite these efforts, while 80% of individuals know that excessive antibiotic use leads to antibiotic resistance [13] and the majority of UK dairy farmers questioned were aware that antibiotic use in livestock can contribute to AR [14], a study in 2018 indicated that approximately 50% of 359 people surveyed still believed that antibiotics are effective against viruses [12]. Furthermore, the same study showed that 75% of pharmacists questioned had been asked to supply antibiotics without a prescription and the majority of these were for a viral illness [12].

Since 2018, public awareness of global health issues has increased in the advent of the COVID-19 pandemic. Whether this health awareness has extended to a greater public understanding of factors that contribute to the emergence of antibiotic resistance and/or better understanding of the magnitude of the risk posed by antibiotic-resistant strains of bacteria is not known. It is also not clear whether there is broad public understanding of the role played by societal, rather than personal factors, in driving AR. Assessing the extent of public knowledge in both the causes and risks of AR is essential to inform effective public information campaigns and to enable effective antibiotic stewardship, given the central role of patients in the management of antibiotic use in particular [12,15].

This study therefore sought to determine public understanding of the causes of increased global AR and the absolute and relative risks posed by antibiotic-resistant bacteria. Our key findings are that there is greater awareness of personal, compared with societal, factors contributing to AR and a correspondingly responsible approach to dealing with excess antibiotics after use. However, participants were ill-informed with regards to appropriate applications of antibiotics and consistently underestimated the global risks posed by the AR crisis.

## Materials and methods

A questionnaire comprising 21 questions was devised using Google Forms to assess public understanding of AR. An online questionnaire was favoured over manual distribution to allow for honest responses and to provide respondents with anonymity. The initial 4 questions sought to gather information relating to the demographics of the participant, a further 9 questions related to personal experience of antibiotics and understanding of correct use/disposal and the remaining 8 questions assessed participant knowledge of the causes and absolute/relative global impact of AR. The majority of the 17 questions relating to AR were multiple choice (13/17) to ensure ready comparison across the cohort and ensure that questions were quick and easy for respondents to answer (S1 Appendix).

The questionnaire did not ask for participant-identifiable information and email addresses of participants were not collected. A participant information sheet and participant agreement form were included at the start of the questionnaire and a compulsory checkbox was used to confirm written participant consent. Participants were therefore unable to continue into the questionnaire without consenting to take part.

The link to access the questionnaire was promoted via the use of social media posts including Facebook, Snapchat and Instagram. The questionnaire opened on 1st July 2020 and closed two weeks later resulting in a total of 106 responses. To increase the number of participants still further, the questionnaire was re-opened on 10th December 2021 for 17 days. Participants were supplied with a copy of the Participant Information Sheet and a Participant Agreement Form. Electronic and anonymous agreement was obtained from all participants. In total 164 respondents completed the questionnaire. Participants were not selected; data relating to patient demographics are shown in accordance with STROBE guidelines [16]. It should be noted however that self selection to this survey may have biased towards those with significant awareness of AR [17].

All data were exported into Microsoft Excel 365 for analysis. Graphs were generated and statistical analyses performed in GraphPad Prism 9. The wordcloud was generated at https:// worditout.com/word-cloud/create using default settings, with only background colour and font altered for clarity. Final graph sizing and export was performed in Adobe Illustrator CC 2019. Statistical analyses used two-way ANOVA (with Tukey's post-hoc test), Fisher's exact test, Chi-squared test, Mann-Whitney U-test and Pearson's correlation as indicated. Data were considered significant if $p < 0.05$.

## Results

Between July 2020 and December 2021, 164 UK-resident adults (self-selected as aged >18 years) responded to an online questionnaire designed to explore attitudes to AR. Respondents identified themselves as belonging to a range of age groups and educational background as shown in Table 1. A bias towards female participants was noted (Table 1; no participants identified as 'transgender' or 'other') and most (125/164) participants were located within the South of England, the remainder being resident in the UK.

Eighty six percent (n = 141) of respondents reported having previously taken a course of antibiotics. Of this group, in the last 5 years 66% (n = 108) had received between 1 and 5 antibiotic prescriptions, with 5% (n = 9) receiving over 5 prescriptions; the remaining respondents

**Table 1. Demographics of respondents.**

| Variable | Number of respondents (%) |
|---|---|
| **Gender** | |
| Male | 63 (38.4) |
| Female | 101 (61.6) |
| **Age range:** | |
| 18–25 | 47 (28.7) |
| 26–35 | 20 (12.2) |
| 36–45 | 24 (14.6) |
| 46–55 | 27 (16.5) |
| 56–65 | 17 (10.4) |
| > 65 | 29 (17.7) |
| **Highest level of formal education** | |
| None | 2 (1.2) |
| GCSE or equivalent | 28 (17.1) |
| A-Levels or equivalent | 52 (31.7) |
| Bachelors degree | 57 (34.8) |
| Masters degree or above | 25 (15.2) |

had not received an antibiotic prescription within this timeframe (S1 Table). Most participants (n = 104; 63%) reported receiving advice on how to take antibiotics (S1 Table).

## Preferential recognition of personal behaviours contributing to AR

To gauge the extent to which respondents understand how antibiotic use and misuse contributes to the threat of AR, volunteers were asked 'Which of the following do you think impact antibiotic resistance (tick all that apply)?'. Six options were provided, all of which contribute to varying degrees to the emergence of antibiotic-resistant bacteria; a seventh option enabled participants to select 'none of the above' (S1 Table and Fig 1). While the majority of individuals (n = 152; 93% of respondents) selected 'over-prescription of antibiotics' as a contributory factor, with the second most selected factor also relating to the behaviour of primary users (patients not finishing their antibiotic course; 65% of respondents), fewer participants recognized the importance of contributory factors related to societal behaviour (e.g. over-use of antibiotics in livestock (50%), absence of new antibiotics being discovered (37%) and poor infection control in healthcare settings (35%)). Furthermore, individuals that selected 'over-prescription of antibiotics' chose fewer additional options compared with those that recognised societal factors, suggesting that individuals that recognise societal factors have a greater general knowledge of contributors to AR (Fig 1).

This trend in preferential recognition of personal behaviour as contributing to AR remained when data were analysed only from those individuals who reported having previously taken a course of antibiotics (n = 141, S1 Fig). Similarly, in answer to the question 'Which of the following are you more concerned about? The amount of antibiotics being prescribed to humans or the amount of antibiotics being prescribed to animals?' nearly twice as many individuals reported concern about the amount of antibiotics used in humans (n = 108, 66%) compared with the amount used in animals (n = 56, 34%; S2 Table).

In line with this apparent awareness of personal responsibility regarding antibiotic use, most respondents reported that they would not stop taking antibiotics if they felt better part-

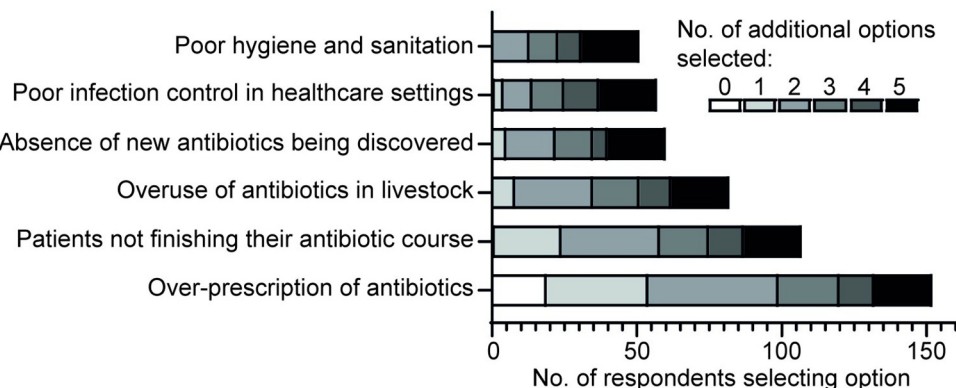

**Fig 1. Factors identified as contributing to AR in response to an on-line questionnaire.** Respondents were asked 'Which of the following factors do you think would impact antibiotic resistance (tick all that apply)?'. Three individuals selected the option 'none of the above' (not shown or included in analysis). The remaining 161 respondents selected one or more of the 6 options as shown. The number of respondents selecting each option is shown, along with an indication of how many additional options were chosen in parallel. Two-way ANOVA p = 0.001 for 'number of additional options' as a factor, p = 0.0007 for each of the contributors to AR as a factor. Significant differences (Tukey's post-hoc test) were observed between the number of additional options selected by those choosing 'over-prescription of antibiotics' compared with those selecting societal factors such as 'poor hygiene and sanitation' (p = 0.0014), 'poor infection control in healthcare settings' (p = 0.0028), 'absence of new antibiotics being discovered' (p = 0.0039) and 'overuse of antibiotics in livestock' (p = 0.04).

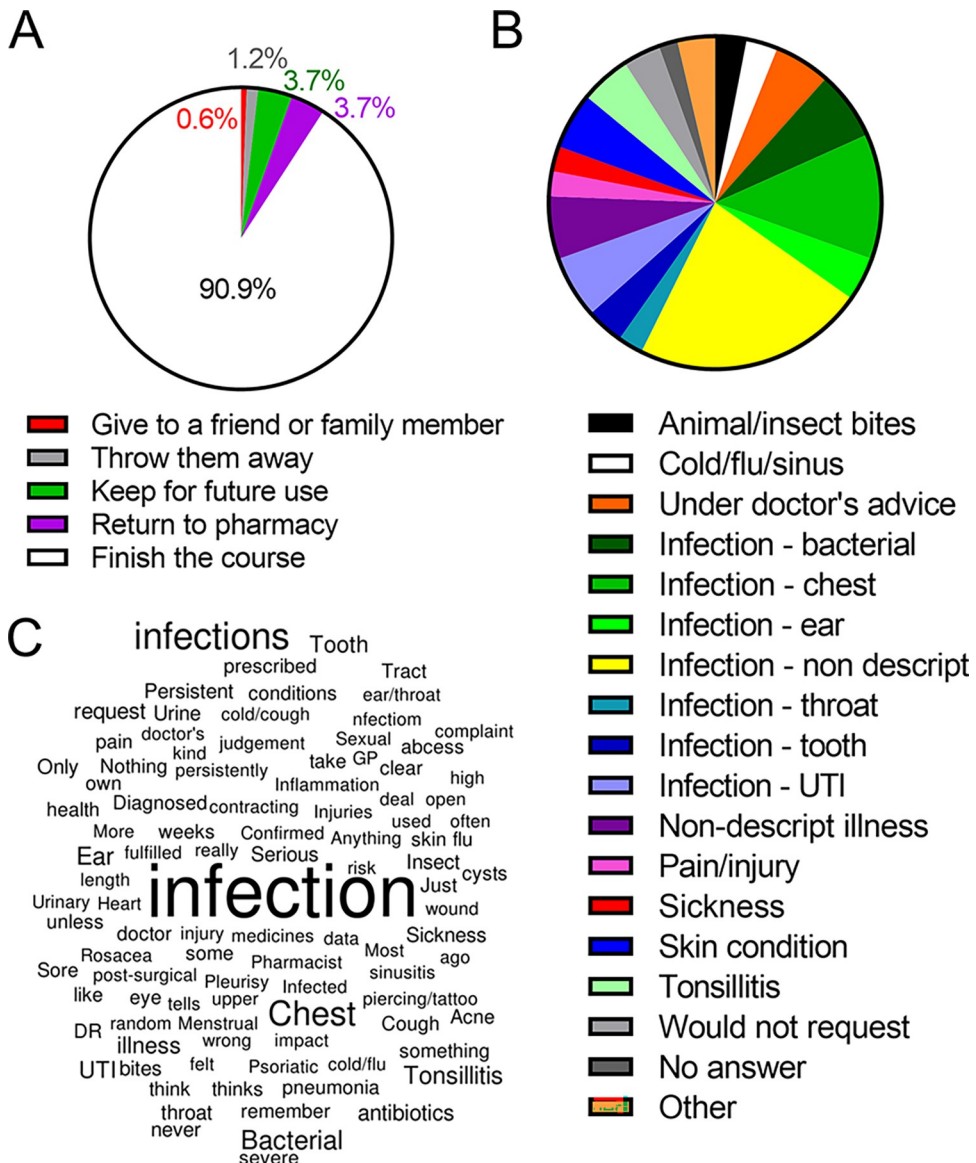

**Fig 2. Public understanding of the need to complete an antibiotic course is not matched by an understanding of when they are most appropriately used.** (A) Respondents were asked 'When prescribed a course of antibiotics, if you felt better halfway through the course, would you stop taking the antibiotics?' and 'If yes, what would you do with the left-over medication?' Four options were provided as answers for the second question as indicated in greyscale above. The percentage of individuals who would (white) or would not (greyscale) complete the course is indicated. (B) Respondents provided a free-text answer to the question 'What are you personally most likely to take antbiotics for (e.g. colds, coughs, infections etc)?'. Responses were grouped into 23 categories as shown above. Six categories (cough/sore throat, depression/anxiety, digestive condition, heart condition, infection–eye, infection-viral) encompassed only 1 respondent each and have been grouped as 'other' for clarity. The % of respondents providing reasons falling within each of the remaining 19 categories was as indicated. (C) Word cloud showing the dominance of 'infection' as a response to the free-text question in (B). (C) Font size reflects frequency of use.

way through a course (n = 149; 91%). Even within the minority of individuals (n = 15) who would stop an antibiotic course part way through, 6 of these would dispose of antibiotics appropriately by returning them to the pharmacy and few participants would consider retaining excess antibiotics for future use (n = 6), throwing them away (n = 2), or giving these to a friend or family member (n = 1) as acceptable (Fig 2A).

Of the whole cohort of 164 respondents, only a minority answered 'yes' to the question 'Do you think it is acceptable to take antibiotics that were given to a friend or family member, as long as they were used to treat the same illness' (n = 23, 14%), again demonstrating an awareness of personal responsibility.

Respondents were also asked to state conditions they would likely request antibiotics for in a free-text format and responses were then grouped for ease of analysis (Fig 2B). Most responses were grouped as 'non-descript infection' (n = 37). In total, 97 respondents stated an infection of some form, yet only 11 respondents (11.3%) specified that this would be for a bacterial infection. The inability of respondents to differentiate between viral and bacterial infection is further implied in some individual responses to this question which included 'flu' (mentioned by 4 respondents), 'cold' (mentioned by a further 2) and 'virus' (mentioned by a further 1 respondent). Furthermore, a total of 20 respondents stated a chest infection without indicating if this was viral or bacterial, casting doubt over whether respondents were aware of the difference.

In addition, non-infective conditions were indicated by individual respondents including 'heart condition', 'rosacea', 'menstrual pain' and 'depression' suggesting a wider lack of understanding of the function of antibiotic medications. Fig 2C shows the word cloud derived from the original and unmodified free-text answers to this question.

## Underestimation of the global impact of antibiotic-resistant bacteria is independent of perceived subject knowledge and educational background

To gain information on respondents' general knowledge of antibiotic resistance, respondents were asked to estimate the current number of deaths attributed to AR worldwide. Options were; under 250,000; 250,000–500,000; 500,001–1,000,000 or over 1,000,000. Forty-eight respondents (29.3%) selected 500,001–1,000,000, which includes the correct estimate of 700,000 [3]. However, responses were spread across the 4 available options suggesting a lack of confidence amongst the cohort (Fig 3A).

A follow-up question asked respondents to grade their perceived knowledge of AR on a subjective scale of 'none', 'some' and 'lots' and separately to provide a free-text answer to the question 'What do you estimate will be the number of deaths worldwide attributed to antibiotic resistance by 2050 according to Public Health England (answer to the nearest million)?'. Most respondents assigned themselves as having 'some' knowledge (72.6%) while 6.7% declared they had 'lots' of knowledge. Those participants who declared no knowledge of antibiotic resistance trended towards an answer closer to the correct value of 10 million compared with those who considered themselves to have 'some' or 'lots' of knowledge (Fig 3B). However, there was no statistically significant difference in the estimates provided by each of the self-declared knowledge groups (p>0.3 for each pair-wise comparison) and a spread in responses was observed.

Similarly, educational background did not influence accuracy of responses to this question (Fig 3C and S2 Table); while there was a trend towards a more accurate answer in those with a degree or higher qualification, compared with those with GCSE or equivalent qualifications as their highest level of education, these apparent differences did not reach statistical significance.

## AR is perceived as a lesser concern than either climate change or cancer, independent of age

To gain insight into respondents' anxiety in relation to antibiotic-resistant bacteria, respondents were asked to rate their relative concern regarding AR compared with other factors of personal/worldwide concern using a series of binary questions ('Which are you more

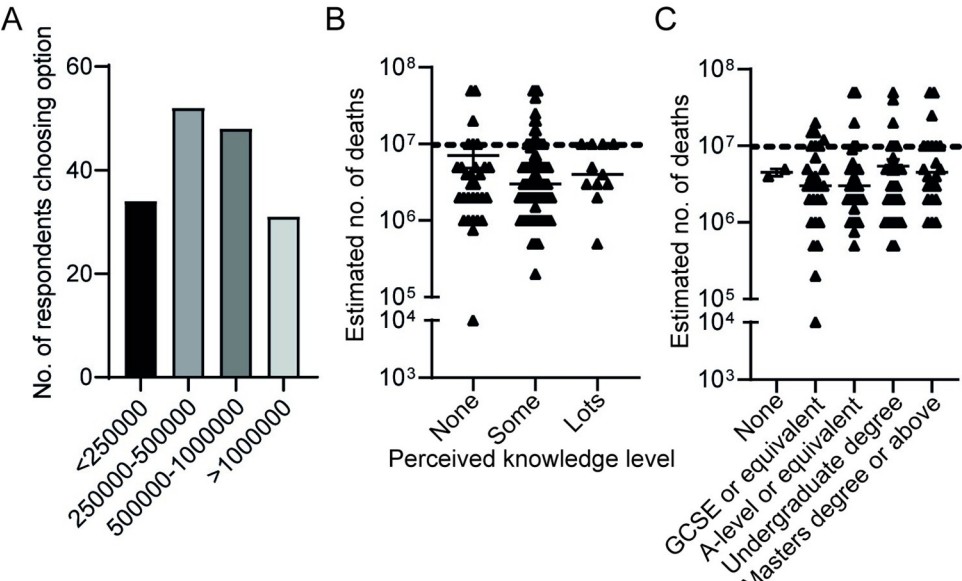

**Fig 3. Actual knowledge of the global impact of AR is independent of perceived knowledge on the subject, or of educational background.** (A) Volunteers were asked to estimate the current number of deaths worldwide due to AR. The number of respondents selecting each of the available options is indicated; all 164 respondents provided an answer to this question. (B) Volunteers were divided into those who considered they had 'none' (n = 34) 'some' (n = 119) or 'lots' (n = 11) of knowledge around AR and their free-text responses to the question 'What do you estimate will be the number of deaths worldwide attributed to antibiotic resistance by 2050, according to Public Health England? Answer to the nearest million' were plotted. Four respondents from the 'none' group and 10 from the 'some' group did not provide a value for this question and their data are not included. (C) Free-text answers to the question in (B) were plotted against the highest educational level reported by each respondent. Seven individuals in the 'A-level or equivalent group', 4 in the 'undergraduate degree' group and 3 in the 'Masters degree or above' group did not provide a value for this question and their data are not included. (B and C) Data points show individual responses with group means +/- SEM indicated. Dashed lines represent the correct estimate of 10 million deaths. Responses were not statistically diferent between groups (Mann-Whitney U tests p>0.05) or between the cohort as a whole (all responders) and the estimate of 10 million (Mann Whitney U test p>0.05).

concerned about; antibiotic resistance or *another factor*'). Factors included in this series of questions were climate change, cancer and diabetes. Respondents showed a clear bias in considering climate change and cancer to be of greater concern than AR (concern ratio for AR (CRAR) of <1; p<0.05 in each case), yet AR trended to be of greater concern than diabetes although this did not reach statistical significance (Fig 4A and S2 Table). The bias towards respondents having greater concern about climate change and cancer but not diabetes relative to AR was largely independent of educational background (S2 Fig and S2 Table).

While cancer and diabetes preferentially affect the elderly [18,19] and may therefore be anticipated as a greater concern than AR in older age groups, climate change is of greater concern for the young [20]. To establish whether the young were biased towards greater concern over climate change compared with AR, and conversely, whether cancer/diabetes were deemed more concerning in the older age groups, responses were grouped into age categories (Fig 4B). While climate change and cancer were both of greater concern than AR in all age groups (CRAR<1), unexpectedly, older age groups trended towards a relatively increased concern about AR compared with cancer (Fig 4B). However, there was no statistical correlation between respondent age and the percentage of respondents selecting AR as greater concern than cancer (S2 Fig). Older individuals trended towards greater concern about AR compared with diabetes, although there was no correlation between age and % of respondents selecting AR as greater concern than diabetes (S2 Table).

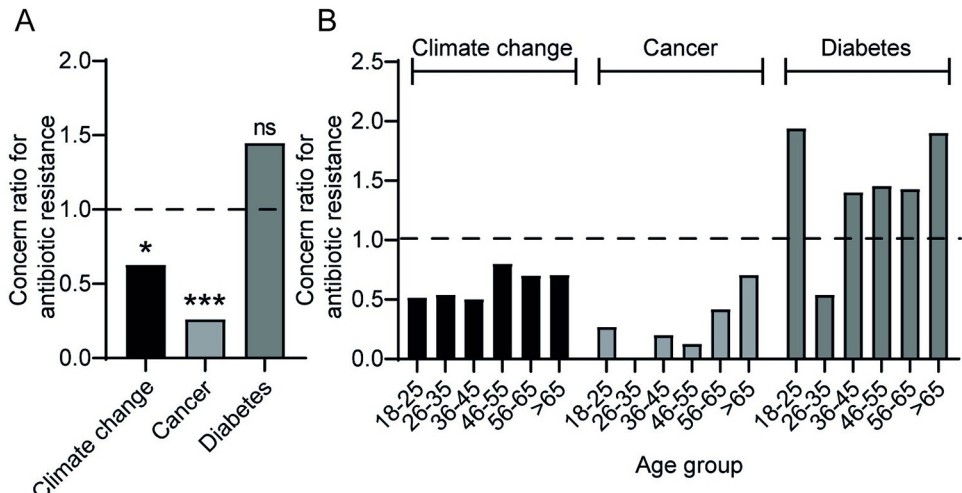

**Fig 4. Climate change and cancer are deemed a greater concern than AR largely independent of age.** (A) Volunteers were asked a series of 3 questions 'Which of the following two options are you more concerned about?' followed by options to choose AR compared with climate change, cancer or diabetes. All respondents provided an answer to these questions. A concern ratio for AR (CRAR) was then generated using the formula 'number of respondents selecting AR/number of respondents selecting alternative factor'. (B) CRAR values were similarly calculated for each age category. (A and B) Values greater than 1 indicate a bias towards concern about AR compared with the alternative factor. (A) Two-sided Fisher's exact test *p = 0.034, ***p<0.0005, ns; p = 0.12.

## Discussion

Minimising the threat of AR requires co-ordinated action by clinicians, veterinary staff and those working in agricultural roles as well as patients and consumers [5,21]. The role played by patients in regulating antibiotic use, and therefore AR, is clear with several studies showing that clinicians expected by the patient to prescribe antibiotics are more likely to do so [22,23]. Furthermore, a significant proportion of patients do not complete a full course of antibiotics, a factor historically associated with increased AR [21,24–27], although this has been called into question in recent years [28,29]. Campaigns aimed at educating the wider public about contributors to AR have been associated with reduced AR rates, yet evaluation of the effectiveness of such campaigns has often not been conducted [21]. Furthermore, few studies have quantified public understanding of wider societal factors that contribute to growing AR rates (e.g. antibiotic use in livestock) yet individual non-healthcare centred decisions to e.g. avoid purchasing meat from animals prophylactically treated with antibiotics, have the potential to further limit the spread of antibiotic-resistant bacteria [30].

We therefore undertook a knowledge, attitudes and practices questionnaire, a component of the World Health Organisation's framework to tackle antimicrobial resistance [13]. This survey was open during the COVID-19 pandemic, when public awareness of health issues was high, yet targeted studies suggest a persistent knowledge gap remains related to appropriate antibiotic use [8,9]. Our data reveal that most respondents recognize over-prescription of antibiotics as contributing to the growth of antibiotic-resistant bacterial strains (>90% of respondents). A study conducted prior to the pandemic report that approximately 82% of respondents answered in the affirmative to a similar set of questions [13]. A systematic review of responses to similar questions indicated 70–74% of individuals equate excessive or unnecessary antibiotic use with the rise of AR and >75% recognize reduction of antibiotic use as an appropriate strategy [31]. The increased % of respondents recognising the importance of over-prescription of antibiotics in the current study may reflect increased public awareness due to

the COVID-19 pandemic, although a larger study would be needed to corroborate this finding. Furthermore, it remains unknown whether this relatively high awareness of the role of over-prescription inversely correlates with the likelihood of respondents to put pressure on health-care providers to prescribe antibiotics. While a question relating to pressurising prescribers was not posed to respondents in this study, and this remains a limitation of the current work, other reports indicate that this is a factor likely to contribute to AR [12,22,23].

Similarly, we find that two thirds of respondents recognized non-completion of an antibiotic course as a factor increasing AR rates and an even higher proportion would themselves complete a course of antibiotics if they felt better halfway through the treatment course. Such high general awareness of the need to complete an antibiotic course in full is consistently found across multiple studies [25,31–33]. Whether this 'complete the course' message needs to change to emphasise the efficacy of short courses of antibiotics and/or completion as pre-scribed, remains controversial [28,29].

In our study only a minority of respondents (4%) stated that they would retain antibiotics for future use or for family/friends, in common with several previous reports [12,13,34]. Inter-estingly, however high frequencies (23–62%) of disparate groups of individuals admit to retaining antibiotics at home [13,25,35]. While UK studies tend to find a lower frequency (5–9%) of respondents admitting to storing antibiotics at home, in line with our findings [12,13], one UK study showed 12% of participants had received antibiotics from friends or family and 7% had taken antibiotics from a previous supply suggesting widespread home storage of anti-biotics within the UK [12].

Relatively few participants in our study were aware of societal impacts such as absence of new antibiotic discoveries (37%) or overuse of antibiotics in livestock (50%), compared with >90% that were aware of over-prescription of antibiotics, as contributors to AR. One Swedish study similarly showed only 45% of participants could identify that antibiotic use for animals can reduce the possibility of effective antibiotic treatment of humans. Akin to our study >90% of these same participants were aware of the role played by excessive antibiotic use in driving AR [33] suggesting that both cohorts were otherwise well informed. A question relating to the absence of new antibiotic discoveries as a driver of AR was not included within the Swedish study. Other studies seeking to define public awareness of wider societal impacts on AR are lacking and further research in this area is needed.

Although the contribution of non-medical antibiotic use to the growth of AR is unclear, antibiotic use in livestock likely exceeds that in humans in developing countries [36]. Further-more, restricting antibiotic use in food-producing animals significantly decreases the presence of antibiotic-resistant bacteria in those animals [37], a finding which led the WHO to recom-mend reduction of all antibiotics in food-producing animals as part of a strategy to reduce AR [38]. Multiple campaigns have sought to encourage reduced antibiotic use by patients and cli-nicians [39], yet more needs to be done to educate the public in their role as consumers, as well as patients, able to contribute to antibiotic stewardship.

One goal of this study was to determine whether increased public awareness of infectious-health medicine in the wake of the COVID-19 pandemic, extends to better appreciation of appropriate personal antibiotic use. A systematic review of surveys conducted largely prior to 2020 showed 29% of participants considered antibiotics useful for treating viral infections [13]. While only 4% of participants in the current study referred to viruses in response to a question about when they might use antibiotics, the absence of responses including the term 'bacteria' or derivatives thereof, was also notable and suggests there remains a lack of understanding. Far more patients with COVID-19 have received antibiotic prescriptions than have had a concur-rent bacterial infection [40]. This finding may reflect caution as regards COVID-19 on the part of health-care professionals as well as pressure imposed on healthcare workers by patients,

given that medical practices with a high rate of antibiotic prescriptions tend towards higher patient satisfaction [41].

Participants consistently under-estimated the current and predicted future risk of AR in the current study, irrespective of their perceived knowledge of the subject or educational background; climate change and cancer were deemed of greater concern than AR. Cancer was responsible for close to 10 million deaths globally in 2020 and 1 in 2 individuals in the UK will be diagnosed with cancer in their lifetime [42,43], potentially exceeding deaths due to AR [4] suggesting good understanding of relative risk by participants. However, by 2050, deaths due to AR are estimated to reach between 10 and 444 million [2–4], matching or far outstripping deaths due to cancer [2]. Based on an average life expectancy in the UK of approximately 80 [44], more than half of participants in this study will be alive in 2050, at a time when their risk of death from AR may be > 40 times greater than that from cancer. This sobering prediction is not reflected in the relative perceived risk of cancer vs AR by participants in this study. Similarly, the WHO predicts that by 2050, climate change is expected to cause approximately 250,000 additional deaths per year, far short of the current 700,000 deaths due to AR per year [45]. This actual relative risk is not reflected by answers given by participants' in the current study who perceived climate change as the greater threat.

Collectively, data from our study reveal that patients are aware of their personal responsibilities regarding antibiotic use for the prevention of AR. However, participants were ill informed as regards the correct clinical use of antibiotics and had relatively poor knowledge of their role as consumers in antibiotic stewardship. Furthermore, perception of AR risk was underestimated using several metrics indicating that further work is needed to engage the public to prevent the looming AR crisis.

## Supporting information

**S1 Fig. Factors identified as contributing to AR by antibiotic-experienced respondents.** Participants who reported that they had previously received antibiotics in the past (n = 141) were asked 'Which of the following factors do you think would impact antibiotic resistance (tick all that apply)?'. The number of respondents selecting each option is shown, along with an indication of how many additional options were chosen in parallel. Two individuals selected the option 'none of the above' (not shown). The remaining 162 respondents selected one or more of the 6 options as shown. Two-way ANOVA p<0.0009 for 'number of additional options' or each of the contributors as a factor. Significant differences (Tukey's post-hoc test) were observed between the number of additional options selected by those choosing 'over-prescription of antibiotics' compared with those selecting societal factors such as 'poor hygiene and sanitation' (p = 0.0016), 'poor infection control in healthcare settings' (p = 0.0027) and 'absence of new antibiotics being discovered' (p = 0.008).
(TIF)

**S2 Fig. Climate change and cancer are deemed a greater concern than AR largely independent of educational background.** (A) CRAR values (as Fig 1) were calculated for AR compared with climate change, cancer and diabetes and responses compared across different educational backgrounds as indicated. Only 2 respondents had an educational background of 'none'; both were more concerned with AR compared with climate change and diabetes but considered cancer more concerning than AR; due to the small number of respondents in these groups, these data were omitted from the graph. Values greater than 1 indicate a bias towards concern about AR compared with the alternative factor. (B) % of respondents in each age group selecting AR as greater concern than cancer. See Table 1 for numbers of individuals in each category; all respondents provided an answer to this question. Pearson's two-tailed

correlation p = 0.14.
(TIF)

**S1 Table. Responses to questions relating to personal use of antibiotics.** [1]Y = Yes, N = No. Numbers indicate number of respondents providing this answer. Numbers in parentheses indicate percentage of group providing this answer. [2]Fisher's exact test, [3]Chi-squared test. [4]Comparing age groups 18–45 with those aged 46 and over. [5]Comparing groups with and without a minimum degree level qualification. NS p>0.05, *p<0.05, **p<0.01.
(DOCX)

**S2 Table. Responses to questions relating to AR risk perception.** [1]Numbers indicate number of respondents providing this answer. Numbers in parentheses indicate percentage of group providing this answer. [2]Full text for this question was "What do you estimate will be the number of deaths worldwide attributed to antibiotic resistance by 2050, according to Public Health England? Answer to the nearest million."Group mean is shown with range in parentheses. [3]Chi-squared test, [4]Mann-Whitney U-test, [5]Fisher's exact test. [6]Comparing age groups 18–45 with those aged 46 and over. [7]Comparing groups with and without a minimum degree level qualification. NS p>0.05, *p<0.05, ***p<0.005.
(DOCX)

**S1 Appendix. Questions included in the on-line questionnaire.**
(DOCX)

## Author Contributions

**Conceptualization:** Emily Smith, Sarah Buchan.

**Data curation:** Emily Smith, Sarah Buchan.

**Formal analysis:** Emily Smith, Sarah Buchan.

**Investigation:** Emily Smith, Sarah Buchan.

**Methodology:** Emily Smith, Sarah Buchan.

**Project administration:** Emily Smith, Sarah Buchan.

**Resources:** Emily Smith.

**Supervision:** Sarah Buchan.

**Visualization:** Emily Smith, Sarah Buchan.

**Writing – original draft:** Sarah Buchan.

**Writing – review & editing:** Emily Smith, Sarah Buchan.

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
