## [Decision Letter · Decision Letter 0]

29 Aug 2023

PONE-D-23-21140Skewed perception of personal behaviour as a contributor to antibiotic resistance and underestimation of the risksPLOS ONE

Dear Dr. Buchan,

Thank you for submitting your manuscript to PLOS ONE. After careful consideration, we feel that it has merit but does not fully meet PLOS ONE’s publication criteria as it currently stands. Therefore, we invite you to submit a revised version of the manuscript that addresses the points raised during the review process. Please ensure that your decision is justified on PLOS ONE’s publication criteria and not, for example, on novelty or perceived impact.

We look forward to receiving your revised manuscript.

Kind regards,

Kshitij Karki, MPH, MA

Academic Editor

PLOS ONE

Additional Editor Comments:

Please respond to the reviewers decision.

Reviewers' comments:

Reviewer's Responses to Questions

**Comments to the Author**

1. Is the manuscript technically sound, and do the data support the conclusions?

Reviewer #1: Yes

Reviewer #2: Yes

2. Has the statistical analysis been performed appropriately and rigorously? 

Reviewer #1: Yes

Reviewer #2: I Don't Know

3. Have the authors made all data underlying the findings in their manuscript fully available?

Reviewer #1: Yes

Reviewer #2: Yes

4. Is the manuscript presented in an intelligible fashion and written in standard English?

Reviewer #1: Yes

Reviewer #2: Yes

5. Review Comments to the Author

Reviewer #1: Good study. The proposal gives incite into the practices in the mentioned geographic area and lays stress on the points which can be acted upon. This will further strengthen the fight against antimicrobial resistance.

Reviewer #2: This is a study of the public’s knowledge, attitudes and habits towards antibiotic use and their perceptions of antibiotic resistance, including personal and societal factors that might affect resistance development. The study also assessed the participants’ perception of the risk of AR relative to other crises such as climate change and cancer, in addition to the absolute risk of AR in terms of the number of deaths now and in the future.

The following comments were noted:

-Please add a copy of the questionnaire as a supplementary file.

-What measures were taken to ensure data confidentiality?

-The questionnaire didn’t assess whether participants would personally put pressure on prescribers to get an antibiotic which would be a personal factor. Please add this to the limitations.

-At times the questions seem to be granular asking about some details such as the projected number of deaths due to AR by 2050 to the nearest million (to be specified not as a choice to be selected from several ranges), whereas another question compares the risk of AR to cancer (neglecting some other important leading causes of death worldwide).

-Did the authors compare the risk of AR to other known leading causes of death such as cardiovascular conditions or just cancer? If the answer is no, please comment on the reason why.

-Line 89: Did the form automatically collect emails of participants?

-Line 112-113: "Electronic and anonymous agreement was obtained from all participants". How was that done? Were these included in the form or as a separate link?

-Line 113: Any reason to explain why the response rate was higher in 2020 relative to 2021? Did the authors do more promotion in 2020? Could it be a heightened public consciousness as a result of COVID in 2020?

-Lines 170-176: Does that mean that participants who recognized over-prescription as a contributor to AR development were less likely to recognize the contribution of other factors?

-Lines 201-203: The percentage is not shown in Figure 2 (A).

-Lines 382-383: 37% is more like one-third of the participants and not just a few participants.

-Line 421: Is that figure right? Half of individuals will be diagnosed with cancer in their lifetime?

6. PLOS authors have the option to publish the peer review history of their article (what does this mean?). If published, this will include your full peer review and any attached files.

Reviewer #1: No

Reviewer #2: **Yes: **Alaa Abouelfetouh

---

## [Author Response · Author response to Decision Letter 0]

11 Sep 2023

Responses to the editor and reviewers are below. Note that all of this information is also included within the 'Response to Reviewer' file uploaded as a separate document. The authors would like to sincerely thank the editor and reviewers for their time in critically evaluating our manuscript. 

Additional Editor Comments:

Please respond to the reviewers decision. Please see responses below.

Reviewers' comments:

Reviewer's Responses to Questions

Comments to the Author

1. Is the manuscript technically sound, and do the data support the conclusions?

Reviewer #1: Yes

Reviewer #2: Yes

2. Has the statistical analysis been performed appropriately and rigorously? 

Reviewer #1: Yes

Reviewer #2: I Don't Know

3. Have the authors made all data underlying the findings in their manuscript fully available?

Reviewer #1: Yes

Reviewer #2: Yes

4. Is the manuscript presented in an intelligible fashion and written in standard English?

Reviewer #1: Yes

Reviewer #2: Yes

We thank the reviewers for their positive responses to questions 1-4. No further comment is required from the authors to these questions above.

5. Review Comments to the Author

Please use the space provided to explain your answers to the questions above. You may also include additional comments for the author, including concerns about dual publication, research ethics, or publication ethics. (Please upload your review as an attachment if it exceeds 20,000 characters).

Our comments are provided below. This information is also supplied in a separate file ‘Response to Reviewers’.

Reviewer #1: Good study. The proposal gives incite into the practices in the mentioned geographic area and lays stress on the points which can be acted upon. This will further strengthen the fight against antimicrobial resistance.

Reviewer #2: This is a study of the public’s knowledge, attitudes and habits towards antibiotic use and their perceptions of antibiotic resistance, including personal and societal factors that might affect resistance development. The study also assessed the participants’ perception of the risk of AR relative to other crises such as climate change and cancer, in addition to the absolute risk of AR in terms of the number of deaths now and in the future.

We thank both reviewers for their comments. Responses to specific points raised are as below. 

-Please add a copy of the questionnaire as a supplementary file.

We apologise that this was not included in the original submission. This omission has now been corrected by inclusion of an additional supplementary file, named ‘S1 Appendix’. We have added reference to S1 Appendix after the description of the questionnaire in the Materials and methods section on line 98. 

-What measures were taken to ensure data confidentiality?

The information provided by each participant was provided anonymously. Respondents completed a Google Forms questionnaire which did not collect any identifying information. For this reason, it is not possible for respondents to be identified and there are therefore no issues of confidentiality for this study. However, and in line with journal requirements, the anonymous data obtained from the study are shared via a repository through the following site: https://doi.org/10.18746/bmth.data.00000316’. 

-The questionnaire didn’t assess whether participants would personally put pressure on prescribers to get an antibiotic which would be a personal factor. Please add this to the limitations.

We thank the reviewer for pointing this out and note that this sort of pressuring behaviour by patients has been noted in other studies (we refer to this in lines 66 and also 334). The reviewer is correct however that this is not something we explored in the current study. To acknowledge this, we have added a short section within the discussion (after line 361 of the original manuscript, corresponding to lines 366-372 in the tracked version) to highlight this as a specific limitation of the study.

-At times the questions seem to be granular asking about some details such as the projected number of deaths due to AR by 2050 to the nearest million (to be specified not as a choice to be selected from several ranges), whereas another question compares the risk of AR to cancer (neglecting some other important leading causes of death worldwide).

-Did the authors compare the risk of AR to other known leading causes of death such as cardiovascular conditions or just cancer? If the answer is no, please comment on the reason why.

The reviewer is correct that some of our questions required a binary response whereas others were more granular. We elected to keep open-text responses to a minimum in this survey (open text answers only comprised 3 of the substantive questions throughout the questionnaire) to facilitate analysis. With regards the second point about comparison of AR with other leading causes of death, we note that the top 5 leading causes of death in the UK in 2022 were, in order, dementia/Alzheimers disease, ischaemic heart disease, chronic lower respiratory disease, cerebrovascular disease and respiratory malignancy* and so the reviewers query about the omission of a question relating to cardiovascular disease is entirely fair. However, we chose to ask ‘which of the following two options are you more worried about?’. While cancer is not objectively the leading cause of death in the UK, given that ½ individuals will develop cancer in their lifetime and cancer is associated with debilitating treatment regimes, it remains a significant ‘worry’ for the public. It is also a term which is broadly understood by non-healthcare professionals, whereas more specific terms such as cardiovascular disease can be misinterpreted by those unclear as to whether stroke or angina for example are encompassed within the term. We were also mindful to avoid bias for conditions which affect a significantly skewed demographic (e.g. Alzheimer’s disease which preferentially affects women). For these reasons, we selected cancer as one of the comparators. A similar rationale fed into our decision to use diabetes as a comparator. We were also influenced in our decisions here by the Review on Antimicrobial Resistance (citation 11 in our manuscript) which selected ‘cancer’ and ‘diabetes’ as their comparisons for predicted AMR rates in 2050 (page 11 of the report). 

 *https://www.ons.gov.uk/peoplepopulationandcommunity/birthsdeathsandmarriages/deaths/articles/deathregistrationsummarystatisticsenglandandwales/2022#:~:text=For%20males%2C%20ischaemic%20heart%20diseases,both%20increases%20compared%20with%202021. 

-Line 89: Did the form automatically collect emails of participants?

Email addresses of participants were not collected by the Google Forms questionnaire. We thank the reviewer for noting that we had not explicitly stated that this was the case however and have added a phrase in line 100-101 to clarify this point.

-Line 112-113: "Electronic and anonymous agreement was obtained from all participants". How was that done? Were these included in the form or as a separate link?

Electronic agreement was obtained from all participants as part of the Google Form itself. The beginning of the questionnaire contained a tick box for participants to agree to take part. This was a compulsory requirement and participants could not progress to the questions until this tick box was checked. To add clarity to the Materials and methods section of the manuscript on this point, we have added the word ‘compulsory’ in line 105 (to now read ‘a compulsory checkbox was used to confirm written participant consent’) and have incorporated an additional sentence starting on line 106 to read ‘Participants were therefore unable to continue into the questionnaire without consenting to take part.’.

-Line 113: Any reason to explain why the response rate was higher in 2020 relative to 2021? Did the authors do more promotion in 2020? Could it be a heightened public consciousness as a result of COVID in 2020?

The reviewer is correct that in 2020 the questionnaire was open for 2 weeks and garnered 106 responses, whereas the subsequent 2021 opening of the same questionnaire (over 17 days) resulted in only 58 responders. I am however mindful that the first responses were obtained in July whereas the second set of responses were obtained in December, prior to Christmas which may have negatively influenced the number of people responding due to competing commitments. Furthermore, July 2020 in the UK was a time of significant social hesitation (there were still local lockdowns)– it’s possible this may have contributed to people having more time available to complete an online questionnaire. Many of these restrictions had eased by December 2021. The extent of promotion by us was similar at both times but due to the complex social factors at play at the time, I hesitate to draw any conclusions about public consciousness. It’s an interesting wider question however and we thank the reviewer for raising it.

-Lines 170-176: Does that mean that participants who recognized over-prescription as a contributor to AR development were less likely to recognize the contribution of other factors?

Compared with those that selected a societal factor (e.g. poor hygiene and sanitation), yes this is correct. The lines referred to here (lines 170-176) are the statistics within the legend to Figure 1. We have tried to convey the biological meaning of these analyses in the text of the manuscript in lines 158-161: ‘Furthermore, individuals selecting contributors not directly related to personal behaviour were more likely to have selected multiple options, indicating greater general knowledge of factors contributing to AR in this cohort.’ However, in response to this comment we have refined the text to try and convey this more clearly. The sentence now reads: ‘Furthermore, individuals that selected ‘over-prescription of antibiotics’ chose fewer additional options compared with those that recognised societal factors, suggesting that individuals that recognise societal factors have a greater general knowledge of contributors to AR (Fig 1)’. This new sentence can be found at lines 160-163 in the tracked version of the manuscript. 

-Lines 201-203: The percentage is not shown in Figure 2 (A).

We thank the reviewer for highlighting this omission. We have now added the % values to Figure 2A. 

-Lines 382-383: 37% is more like one-third of the participants and not just a few participants.

We thank the reviewer for pointing out this sentence is misleading. The point that we were trying to make is that fewer participants were aware of societal contributors to antibiotic resistance compared with personal contributors. However, we appreciate that the sentence as written does not convey this clearly. We have therefore amended the sentence improve clarity. The sentence now reads ‘Relatively few participants in our study were aware of societal impacts such as absence of new antibiotic discoveries (37%) or overuse of antibiotics in livestock (50%), compared with >90% that were aware of over-prescription of antibiotics, as contributors to AR.’ This amended sentence can be found at lines 393-396 in the tracked version of the document.

-Line 421: Is that figure right? Half of individuals will be diagnosed with cancer in their lifetime?

We thank the reviewer for this question. The figure is correct but on revisiting the sentence we realise that a citation for this specific point was not included and in response new citation 43 has been added (and subsequent citation numbers adjusted accordingly). This statistic comes from Cancer Research UK, with the data ultimately taken from the national cancer registry of the UK. We have however amended this sentence (lines 421-423 in the original manuscript corresponding to lines 433-435 in the tracked version) to better reflect that these data are from the UK rather than worldwide.

---

## [Decision Letter · Decision Letter 1]

8 Oct 2023

Skewed perception of personal behaviour as a contributor to antibiotic resistance and underestimation of the risks

PONE-D-23-21140R1

Dear Dr. Sarah Buchan,

We’re pleased to inform you that your manuscript has been judged scientifically suitable for publication and will be formally accepted for publication once it meets all outstanding technical requirements.

Kind regards,

Kshitij Karki, MPH, MA

Academic Editor

PLOS ONE

Additional Editor Comments (optional):

Reviewers' comments:

Reviewer's Responses to Questions

**Comments to the Author**

1. If the authors have adequately addressed your comments raised in a previous round of review and you feel that this manuscript is now acceptable for publication, you may indicate that here to bypass the “Comments to the Author” section, enter your conflict of interest statement in the “Confidential to Editor” section, and submit your "Accept" recommendation.

Reviewer #2: All comments have been addressed

2. Is the manuscript technically sound, and do the data support the conclusions?

Reviewer #2: (No Response)

3. Has the statistical analysis been performed appropriately and rigorously? 

Reviewer #2: (No Response)

4. Have the authors made all data underlying the findings in their manuscript fully available?

Reviewer #2: (No Response)

5. Is the manuscript presented in an intelligible fashion and written in standard English?

Reviewer #2: (No Response)

6. Review Comments to the Author

Reviewer #2: (No Response)

7. PLOS authors have the option to publish the peer review history of their article (what does this mean?). If published, this will include your full peer review and any attached files.

Reviewer #2: **Yes: **Alaa Abouelfetouh

---

## [Editor Report · Acceptance letter]

25 Oct 2023

PONE-D-23-21140R1 

Skewed perception of personal behaviour as a contributor to antibiotic resistance and underestimation of the risks 

Dear Dr. Buchan:

I'm pleased to inform you that your manuscript has been deemed suitable for publication in PLOS ONE. Congratulations! Your manuscript is now with our production department. 

Kind regards, 

on behalf of

Dr. Kshitij Karki 

Academic Editor

PLOS ONE